# Strategies Tackling Viral Replication and Inflammatory Pathways as Early Pharmacological Treatment for SARS-CoV-2 Infection: Any Potential Role for Ketoprofen Lysine Salt?

**DOI:** 10.3390/molecules27248919

**Published:** 2022-12-15

**Authors:** Domenica Francesca Mariniello, Valentino Allocca, Vito D’Agnano, Riccardo Villaro, Luigi Lanata, Michela Bagnasco, Luigi Aronne, Andrea Bianco, Fabio Perrotta

**Affiliations:** 1Department of Translational Medical Sciences, University of Campania “L. Vanvitelli”, 80131 Naples, Italy; 2U.O.C. Clinica Pneumologica “L. Vanvitelli”, A.O. dei Colli, Ospedale Monaldi, 80131 Naples, Italy; 3Section of Infectious Diseases, Department of Clinical Medicine and Surgery, University of Naples Federico II, 80131 Naples, Italy; 4Medical Deptartment, Dompé Farmaceutici SpA, 20122 Milan, Italy

**Keywords:** NSAIDs, COVID-19, SARS-CoV-2, inflammation, ketoprofen

## Abstract

COVID-19 is an infective disease resulting in widespread respiratory and non-respiratory symptoms prompted by SARS-CoV-2 infection. Interaction between SARS-CoV-2 and host cell receptors prompts activation of pro-inflammatory pathways which are involved in epithelial and endothelial damage mechanisms even after viral clearance. Since inflammation has been recognized as a critical step in COVID-19, anti-inflammatory therapies, including both steroids and non-steroids as well as cytokine inhibitors, have been proposed. Early treatment of COVID-19 has the potential to affect the clinical course of the disease regardless of underlying comorbid conditions. Non-steroidal anti-inflammatory drugs (NSAIDs), which are widely used for symptomatic relief of upper airway infections, became the mainstay of early phase treatment of COVID-19. In this review, we discuss the current evidence for using NSAIDs in early phases of SARS-CoV-2 infection with focus on ketoprofen lysine salt based on its pharmacodynamic and pharmacokinetic features.

## 1. Introduction

The severe acute respiratory syndrome coronavirus 2 (SARS-CoV-2), responsible for the novel coronavirus disease 2019 (COVID-19), has been rapidly transmitted around the world during the last three years, causing a global public health emergency [1]. Coronaviruses are pathogens that largely affect the respiratory system but the expression of host SARS-CoV-2 receptor, angiotensin-converting enzyme 2 (ACE2), is not lung-specific, and its presence in a variety of tissues, including the brain, the intestine, the blood vessels, and the kidney, could subject these organs to direct infection by SARS-CoV-2 making COVID-19 a systemic disease [2]. Most patients commonly present with fever, myalgia, shortness of breath, malaise, and dry cough, although patients may present with asymptomatic, mild, moderate, or severe disease. High systemic levels of cytokines, referred to as “cytokine storm”, have frequently been found in severe COVID-19 disease. Therefore, targeting inflammation is one of the key strategies in the management of COVID-19 disease. In this scenario, recent data suggest that NSAIDs may represent a safe strategy in the treatment of SARS-CoV-2 infection [3].

### SARS-CoV-2 Pathogenesis and Associated Damage Mechanisms

SARS-CoV-2 enters the human body via the upper airways. In concordance with other coronaviruses, it is transmitted mainly through respiratory droplets, although other ways of transmission are reported such as aerosol, contact with contaminated surfaces, and fecal–oral transmission [4]. Initially, the virus proliferates in the upper respiratory tract and then the virus moves into the lower respiratory airway. The virus is made up of six accessory proteins and four main structural proteins: nucleocapsid (N), membrane (M), envelope (E), and spike (S), the latter one responsible for viral entry [5]. SARS-CoV2 binds to the host receptor membrane angiotensin-converting enzyme 2 (ACE2) [6,7]. To enable viral entry, the spike (S) protein is cleaved into two subunits, S1 and S2; the S1 subunit is responsible for binding to ACE2 [8]. Type 2 transmembrane cellular proteases (TMPRSS2) mediate the proteolytic cleavage of S-protein [9]. The viral genome is translated by host machinery into a polyprotein, virions are assembled and then released by exocytosis, ready to infect other cells. After viral entry, ACE2 receptor is internalized and degraded [10]. Within a few hours of infection, the recognition of viral components called pathogen-associated molecular patterns (PAMPs) by pattern recognition receptors (PRRs) in the alveolar cells and site of invasion, such as Toll-like receptors and retinoic acid-inducible gene-1 (RIG-1)-like receptors, activate the innate immune response [11]. Nuclear factor kB (NF-κB) and interferon regulatory factor 3 (IRF3) initiate the expression of interferon type 1 and cytokines such as tumor necrosis factor alpha (TNF-a), interleukin-1 (IL-1), and IL-6. These cytokines represent the first line of host defense and subsequently induce adaptive immune responses [12]. The upregulation of the NF-κB signaling pathway leads to cytokine storm and hyperinflammation, with a greater risk of severe COVID-19 disease [13,14]. Regarding adaptive immune response, T lymphocytes attempt to eliminate virus by killing infected cells and secrete cytokines to amplify T lymphocytes immune response [15]. Severe COVID-19 patients often exhibit lymphopenia with a reduction in the number of T-cells, probably due to chemotaxis of these cells toward the site of inflammation or to direct cell damage provoked by SARS-CoV-2 [16].

B lymphocytes produce antibodies specific to SARS-CoV-2, in particular antibodies versus spike protein, which may neutralize the virus and stimulate systemic immunity. Reduction in the B cell count has been observed in patients with severe COVID-19.

Early autoptic studies documented in patients deceased from COVID-19 diffuse pulmonary microthrombi that appear to form directly within pulmonary vessels and do not result from embolic propagation of peripheral venous thrombi [17,18,19]. In severe COVID-19 patients, reduction in platelet count with augmentation of fibrinogen and dimer levels is commonly observed [1]. There are several molecular/cellular pathways that potentially can explain this hypercoagulable state in COVID-19: it may involve ACE2, a carboxypeptidase responsible for the conversion of angiotensin II (Ang II) to angiotensin 1-7. As mentioned above, after binding with the spike protein of the virus, ACE2 is internalized and degraded [20], leading to up-regulation of Ang II signaling and consequentially of pro-thrombotic pathways. On the other hand, the level of angiotensin 1-7 is diminished [21]. Angiotensin 1-7 mediates an anti-thrombotic activity by the production of nitric oxide and prostacyclin and the inhibition of platelet activation [22]. In addition, the hyperinflammation in COVID-19 disease mediated by cytokines such as IL-1, TNF-α, and IL-6 also leads to a hypofibrinolytic state through an increase of fibrinogen and plasminogen activator inhibitor-1 (PAI-1) [23]. Inflammatory cytokines promote the activation and aggregation of platelets, the activation of vascular endothelial cells (EC) and the expression of tissue factor (TF) on the surface of endothelium cells and leucocytes [24]. Furthermore, during inflammation, the production of natural anticoagulants such as antithrombin III, tissue factor inhibitor, and Protein C is diminished [25]. A vicious circle is established as the coagulation cascade can in turn promote inflammation. In fact, thrombin is a major activator of protease-activated receptor 1 (PAR 1) that contributes to the release of IL-1, IL-2, IL-6, IL-8, TNFα and the expression of adhesion molecules such as E- and P-selectin and ICAM-1 on the endothelial surface involved in immune cell recruitment [26]. This review aims to summarize the current therapeutic scenario for the early phases of COVID-19 focusing on the role of anti-inflammatory treatment that appears still controversial.

## 2. Pharmacological Agents with Selective Activity against SARS-CoV-2

### 2.1. Antivirals Targeting SARS-CoV-2

The group of antiviral drugs comprises several molecules directly targeting the pathogen to hinder its growth. According to their different mechanisms of action, antiviral drugs may be categorized into three main subgroups. (1) Inhibitors of S protein; (2) inhibitors of viral proteases; (3) inhibitors of viral RNA dependent RNA polymerase; (4) host-oriented.

#### 2.1.1. Entry Inhibitors

As stated above, S protein is responsible for virus entry. The S protein is a transmembrane protein with N-exo and C-endo terminals. The N terminal S1 subunit contains receptor binding domain (RBD), known to interact with the peptidase domain of ACE2 and to be the main target of neutralizing antibodies, while the C terminal S2 subunit induces membrane fusion. Monoclonal antibodies (mAbs) are laboratory-produced molecules which derive from natural B cells of subjects who have experienced or been injected with the antigen of interest. As a result, mAbs are able to mimic a normal immune response against a predetermined antigen [27]. During SARS-CoV-2 pandemic, various mABs have been progressively approved worldwide, whilst other are currently under investigation [28]. Target population for treatment with Abs is represented by high-risk patients with symptomatic mild to moderate infection, not requiring supplemental oxygen due to COVID-19. High risk features, among others, include older age, chronic kidney disease, obesity, cardiovascular and metabolic disease, and chronic lung diseases.

##### Bamlanivimab-Etesevimab

In the BLAZE-1 trial (*NCT04427501*), the cocktail of bamlanivimab and etesevimab, has been reported to significantly reduce hospitalizations and death rate in high-risk patients affected by COVID-19. At the end of 29 days observational period, in the bamlanivimab-etesevimab group, a total of 11 of 518 patients (2.1%) experienced COVID-19 related hospitalization or death from any cause, as compared with 36 of 517 patients (7.0%) in the placebo group (absolute risk difference, −4.8 percentage points; 95% confidence interval [CI], −7.4 to −2.3; relative risk difference, 70%; *p* < 0.001). Contrary to placebo group, where nine of the ten occurred deaths have been designed as COVID-19 related, no patients died in the bamlanivimab-etesevimab group [29].

##### Casirivimab and Imdevimab

REGN-COV2 (casirivimab and imdevimab; *NCT04452318*) represents a mixture of the human Abs, casirivimab, and imdevimab. Although identified by different methods, they both target the S protein RBD. Subcutaneous casirivimab and imdevimab, 1200 mg, succeeded in preventing progression to symptomatic disease compared to placebo; odds ratio, 0.54 [95% CI, 0.30–0.97]; *p* = 0.04; absolute risk difference, −13.3% [95% CI, −26.3% to −0.3%]). Furthermore, the combination of casirivimab and imdevimab has been reported to reduce the number of symptomatic weeks per 1000 participants (895.7 weeks vs. 1637.4 weeks with placebo; *p* = 0.03) with an approximately 5.6-day reduction in symptom duration per symptomatic participant. With respect to adverse events, the proportion of participants receiving casirivimab and imdevimab who experienced one or more treatment-emergent adverse events was 33.5% compared to 48.1% for placebo, including events related (25.8% vs. 39.7%) or not related (11.0% vs. 16.0%) to COVID-19 [30].

##### Sotrovimab

Sotrovimab, a pan-sarbecovirus monoclonal antibody was shown to significantly reduce the risk of disease progression, leading to hospitalization (for >24 h) for any cause or death, as demonstrated in a phase 3, multicenter, randomized, doubleblind, placebo-controlled trial. A total of 3 of 291 patients in the sotrovimab group (1%) experienced hospitalization for >24 h for any cause or death, in comparison to 21 of 292 patients in the placebo group (7%) (relative risk reduction 85%; 97.24% confidence interval [CI], 44 to 96; *p* = 0.002) [31].

##### Tixagevimab-Cilgavimab

Tixagevimab-cilgavimab is a neutralizing monoclonal antibody combination whose capability to improve outcomes for patients hospitalized with COVID-19 has lately been investigated. Although tixagevimab–cilgavimab did not improve the primary outcome of time to sustained recovery versus placebo, (89% for tixagevimab–cilgavimab and 86% for placebo group at day 90 [(recovery rate ratio [RRR] 1.08 [95% CI 0.97–1.20]; *p* = 0.21), it must be noted that mortality was lower in the tixagevimab–cilgavimab group (61 [9%]) versus placebo group (86 [12%]; hazard ratio [HR] 0.70 [95% CI 0.50–0.97]; *p* = 0.032) [32].

##### Other Anti-SARS-CoV-2 Monoclonal Antibodies

SARS-CoV-2 may mutate over time making certain treatments less useful and allowing the pandemic to spread. In this respect, variants of concern (VOCs) are continuously monitored due to their great impact on decreasing efficacy of treatment with mAbs. As omicron VOC has quickly spread becoming the dominant variant in US, bebtelovimab has been considered the only monoclonal antibody-based treatment approved by the Food and Drug Administration (FDA) at present [33]. Based on its capabilities to link S protein amino acids that are reported to be rarely mutated, bebtelovimab, a fully human immunoglobulin G1, may represent a long-term solution for COVID-19 treatment as suggested by Westendorf K et al. [34].

#### 2.1.2. Inhibitors of Viral Proteases

SARS-CoV-2 is constituted by four conserved structural proteins—spike (S), envelope (E), membrane (M), and nucleocapsid (N)—and six accessory proteins. Among them, there are two recognized cysteine proteases—Mpro (3CLpro) and PLpro—which are essential for viral replication. Given the absence of human homolog as well as its important role in the viral gene expression, Mpro has been utilized as a potential molecular target.

##### Lopinavir/Ritonavir

Lopinavir is an anti-retroviral protease inhibitor employed in combination with ritonavir, in the treatment of HIV infection. Although lopinavir/ritonavir administration was not associated with a significant difference in the time to clinical improvement, in a post-hoc analysis, 28-day mortality was lower in treated population compared to control, albeit not significantly (19.2% vs. 25%) [35]. At present, lopinavir/ritonavir is not recommended for COVID-19 treatment and it can only be considered for patients included in clinical trials [33,36].

##### Nirmatrelvir/Ritonavir

Nirmatrelvir, a novel orally active inhibitor of 3CL protease inhibitor, in combination with ritonavir, has been investigated in a phase III trial in a cohort of 2246 symptomatic, unvaccinated, non-hospitalized adults at high risk for progression. Authors reported that the incidence of COVID-19-related hospitalization or death by day 28 was 0.77% (3 of 389 patients) in the nirmatrelvir group compared to 7.01% (27 of 385 patients) in the placebo group, with 7 deaths, and a 89.1% relative risk reduction. Results were confirmed in the final analysis involving the 1379 patients with a difference in terms of hospitalization of −5.81 percentage points (95% CI, −7.78 to −3.84; *p* < 0.001; relative risk reduction, 88.9%) [34]. Currently, ritonavir boosted-nirmatrelvir is recommended within 5 days of symptoms onset, for non-hospitalized adults and pediatric patients aged ≥12 years and weighing ≥40 kg affected by mild to moderate COVID-19 with high risk of disease progression [33].

#### 2.1.3. Inhibitors of Viral RNA Dependent RNA Polymerase (RdRp)

##### Remdesivir

Originally developed for the treatment of Ebola and Marburg virus infections, remdesivir (GS-5734) was considered early on as a potential candidate for COVID-19 treatment due to its capacity to cause premature termination of SARS-CoV-2 viral RNA transcription. By acting as nucleotide analog, it is incorporated by the RdRp, and RNA synthesis is consequently inhibited. In the Adaptive COVID-19 Treatment Trial (ACTT-1), in a cohort of 1062 randomized patients, subjects who received remdesivir had a median recovery time of 10 days compared to 15 days of the control group (rate ratio for recovery, 1.29; 95% CI, 1.12 to 1.49; *p* < 0.001, by a log-rank test). With regard to mortality, remdesivir has shown superiority compared to placebo both at day 15 and day 29 (6.7% and 11.4% with remdesivir vs. 11.9 and 15.2% in control, respectively; hazard ratio, 0.73; 95% CI, 0.52 to 1.03). Currently, remdesivir has been recommended for hospitalized patients who require (BIIa) or do not require (BIII) conventional oxygen supplementation. Likewise, the use of remdesivir might be considered for hospitalized patients who require High Flow Nasal Cannula (HFNC) Oxygen or Non-Invasive Ventilation (NIV) in association with dexamethasone plus per os (PO) baricitinib (AI) or IV tocilizumab (BIIa) [37]. Remdesivir has also been recommended for nonhospitalized patients with mild to moderate COVID-19 who were at high risk of progression to severe disease.

##### Molnupiravir

Molnupiravir, the biological prodrug of NHC (β-D-N(4)-hydroxycytidine), represents another ribonucleoside analogue with activity against SARS-CoV-2 and other RNA viruses. Started within 5 days of the onset of signs or symptoms, molnupiravir has showed superiority in decreasing the risk of hospitalization for any cause or death through to day 29 compared to controls in a cohort of 1433 nonhospitalized adults with mild-to-moderate COVID-19 with at least one risk factor for severe COVID-19 (28 of 385 participants [7.3%]) than with placebo (53 of 377 [14.1%]) (difference, −6.8 percentage points; 95% confidence interval [CI], −11.3 to −2.4; *p* = 0.001) [38]. On December 2021, the Food and Drug Administration (FDA) approved monlupiravir for the treatment of adults with mild or moderate COVID-19, within five days of symptom onset, with no alternative antiviral therapies available.

#### 2.1.4. Host-Oriented Therapies for SARS-CoV-2 Infection

Novel drugs targeting host immune factors named host-oriented therapies are currently under development. Virus-host cell interaction prompts the innate immune system response which may be ineffective in determining a complete viral clearance [39,40]. Therefore, the possibility to restore the altered immune responses with these host-oriented therapies offers a great opportunity against viral infections [41,42,43]. SNG001 is an inhaled drug containing INF-β, an antiviral protein produced during viral spread and it is under evaluation in clinical trial (NCT04385095). SARS-CoV-2 might weaken the immune system response also through the inhibition of IFN-β expression [44,45]. In this phase 2 trial, COVID-19 patients receiving inhaled SNG001 had greater improvement in clinical symptoms and recovered more rapidly than patients who received placebo [45]. Another host-oriented strategy in evaluation for COVID-19 is the possibility to use IL-7 to support the host’s immune system. IL-7 promotes lymphocytic count increase counteracting the lymphocytopenia, a pathologic hallmark of severe COVID-19. In a recent study with a small number of patients with COVID-19, it was shown that IL-7 can be safely administered and it was associated with an increase in lymphocytes count, appearing to counteract a pathologic hallmark of COVID-19 [46]. These results were under evaluation in another trial (NCT04379076), actually terminated for poor accrual. The administration of IL-7 seems to improve clinical outcome as already demonstrated in septic patients [47,48]. However, for the lack of strong evidence to support the use of IL-7 in COVID-19 patients, more studies are needed to approve its clinical use. Other drugs which directly affect viral entry are under evaluation. Aprotinin, a serine protease inhibitor that could inhibit TMPRSS2, seems to have potential role in the control of SARS-CoV-2 replication, especially in early stages, and in the prevention of COVID-19 progression to a severe disease [47]. Host factors assisting in viral replication are other targets for host-oriented therapies [49]. Signaling via tyrosine kinases play a crucial role in viral replications and receptor tyrosine kinase inhibitors such as genistein could have a role to treat COVID-19 [50]. Genistein is an isoflavone with potent anti-inflammatory and immunomodulators effects due to its ability to modulate intracellular pathways such as PI3K, Akt, mTOR, NF-κB, PPARγ, AMPK, and Nrf2, preventing viral entry and therefore reducing lung injuries [42,50]. Host-oriented therapies also could be used to mitigate the cytokine storm induced by SARS-CoV-2 and other viral infection due hyper-activation of TLR. Several TLR-4 antagonists such as eritoran are under evaluation for its use in viral infection for their ability to mitigate host damage due to excessive inflammation [51].

## 3. Anti-Inflammatory Drugs in COVID-19

### 3.1. Corticosteroids Use in COVID-19 Patients

Corticosteroids (CCS) are steroid hormones implicated in several physiological processes such as the control of inflammatory response, protein catabolism, gluconeogenesis, antiallergy proprieties, and potent immunomodulator effects [52]. In clinical practice, there are two major classes of corticosteroids that are commonly used, glucocorticoids (e.g., dexamethasone, prednisolone and methylprednisolone) named for their gluconeogenic proprieties, and mineralcorticoids (e.g., fludrocortisone) named for their role in salt-water balance [53]. The potent anti-inflammatory effect of these drugs is due to the inhibition of NF-κB pathway, reducing IL-6 and TNF-α expression. Corticosteroids also act in cellular immunity inhibiting CD8^+^ T cells, TH1 cells, and NK cells [54,55].

Despite these proprieties, the potential side effects such as risk of secondary infection, long-term complications, and delayed viral clearance initially discouraged the use of systemic corticosteroids in COVID-19 patients [54]. Subsequently, the RECOVERY trial, published in 2021, an open-label trial comparing a range of possible treatments in patients who were hospitalized with COVID-19, has demonstrated that the use of dexamethasone 6 mg reduced 28-days mortality in hospitalized patients who were receiving ventilatory support or oxygen alone but not in patients who were receiving no respiratory support [55]. The role of corticosteroids is controversial in mild-moderate COVID-19 and their use in the early stage of the disease. A systematic review showed that patients with mild-moderate COVID-19 treated with CCS have a longer hospitalization and more days of viral shedding [56]. In another randomized controlled trial (RCT) of adult patients with acute hypoxemic failure related to SARS-CoV-2 infection who received corticosteroids versus any comparator, the use of corticosteroids increased mortality in the subgroup of patients not requiring respiratory support [57]. In addition, the National Institutes of Health (NIH) guidelines for the treatment of COVID-19 established that there are no data to support the use of systemic corticosteroids in non-hospitalized patients with COVID-19 [34] and Infectious Diseases Society of America (IDSA) guidelines stated that there was no evidence for benefit with dexamethasone in patients who were not on supplemental oxygen [34]. Another study on the use of dexamethasone especially in early stage of disease appears to show it to be associated with severe COVID-19 due, probably, to an increase in viral load [58]. The increase of mortality related to the early use of corticosteroids is shown in the Liu and Zhang study that showed that patients who received corticosteroid therapy early have poorer outcomes due to a delayed viral clearance [59,60]. For the use in the early phase of SARS-CoV-2 infection, inhaled corticosteroids are being tested in several RCTs in the subset of patients with mild-moderate disease [54]. In the OIC (Steroid in COVID) trial, budesonide dry powder, administered at dose of 800 mcg twice daily (bid) within 7 days of onset of mild symptom, has demonstrated a reduction in urgent medical care and reduced time of recovery with low rate of adverse events [61]. These data were confirmed in the largest PRINCIPLE trial that investigated the use of inhaled budesonide (800 mcg bid) in outpatients aged 65 years old or 50 years old with comorbidities within 14 days of symptoms onset. This study has demonstrated that inhaled budesonide improves the time of recovery reducing the probability of hospital admissions or deaths in people who are at higher risk of complications although the combined endpoint was not achieved [62].

These benefits are not shown with all inhaled corticosteroids. The COVERAGE study, which tested inhaled ciclesonide in 217 outpatients with COVID-19, risk factors for severe disease, symptoms onset ≤ 7 days, and no criteria for hospitalization, revealed no efficacy to reducing the need for oxygen therapy, hospitalization, and/or death [63]. Another study investigated the role of a combination of inhaled ciclesonide and intranasal ciclesonide in young patients with COVID-19 and could not show a statistical improvement in respiratory symptoms compared to the control group [64]. Inhaled steroids are relatively safe drugs with a low risk of side-effects, but their utility in the early stage of COVID-19 requires further studies to better define their role [54].

### 3.2. Non-Steroidal Anti-Inflammatory Drugs in the Early Stage of the Therapeutic Scenario of COVID-19

Non-steroidal anti-inflammatory drugs (NSAIDs) are very widely used to alleviate fever, pain, and inflammation (common symptoms of COVID-19 patients) through effectively blocking production of prostaglandins (PGs) via inhibition of cyclooxygenase enzymes, namely COX-1 and COX-2, that catalyze the two-step conversion of arachidonic acid into thromboxane, prostaglandins, and prostacyclins. Prostaglandins are key inflammatory mediators. The use of NSAIDs during the COVID-19 pandemic, especially in the first wave, was controversial and NSAIDs were largely avoided in this phase of COVID-19 pandemic to favor the analgesic antipyretic paracetamol with no anti-inflammatory effect. Later, on the 18th of March 2020 European Medicines Agency (EMA) clarified and concluded that there is no clinical reason to withdraw the use of NSAIDs during the SARS-CoV-2 pandemic [65]. Moreover, some evidence suggested that early use of non-steroidal anti-inflammatory drugs in COVID-19 might interfere with the disease progression in patients with mild-to-moderate COVID-19 [66,67]. Similarly, in a Danish cohort study, the use of NSAIDs in subjects positive for SARS-CoV-2 was not associated with 30-day mortality, hospitalization, or complications [68]. Non-steroidal anti-inflammatory drugs (NSAIDs) form a large heterogeneous pharmacological family which inhibit COX-1 or COX-2; COX-1, constitutively expressed in the body, plays a pivotal role in platelet aggregation, gastrointestinal barrier integrity, and maintenance of renal function, while COX-2 is expressed during an inflammatory response where pro-inflammatory cytokines or growth factors stimulation are produced, thus being considered the most relevant mediator in promoting inflammation, fever, and pain. In particular, in SARS-CoV-2 infection it has been demonstrated that proteins S and N of the viral nucleocapsid induce the overexpression of COX-2, responsible for the progression of the inflammatory storm, as well as of the disease [69,70].

NSAIDs are defined nonselective when they inhibit both COX-1 and COX-2, and COX-2 selective. Their main indications are to alleviate fever, inflammation, and pain, including in patients with chronic inflammatory disorders such as rheumatoid arthritis and osteoarthritis. To reduce gastrointestinal side effects mediated essentially by inhibition of COX-1, highly selective COX-2 inhibitors (-coxib) were developed, although a number of recent studies have highlighted more serious potential cardiovascular side effects. The cardiovascular risk is not limited to the use of COX-2 inhibitors but is also correlated with the use of non-selective NSAIDs.

It is now recognized that an excessive inflammation is a key feature of severe COVID-19. Since PGs contribute to strengthen the inflammatory cytokines production, NSAIDs may mitigate this pathologic feature. As demonstrated by Chen and co-workers, although NSAIDs did not influence the expression of ACE2 and did not affect both SARS-CoV-2 entry and replication, meloxicam dampened the production of a subgroup of proinflammatory cytokines induced by infection, such as IL-6, CCL2, GM-CSF, CXCL10, IL-2, and TNF-α [71]. Moreover, NSAIDs may promote a more efficient virus clearance through inhibition of PGD_2_ and PGE_2_, which have been reported to impair both innate and adaptive immunity [72,73]. On the other hand, a reduced production of PGI_2_ may be potentially detrimental considering its capacity of modulate virally induced illness [74]. In a recent metanalysis, the risk of severe COVID-19 has been demonstrated to be significantly decreased in subject receiving NSAIDs before admission (adjusted OR = 0.79, 95% CI 0.71–0.89, I2 = 0%) as well as the risk of death compared to controls (adjusted OR = 0.68, 95% CI 0.52–0.89, I2 = 85%) [75]. However, NSAIDs usage is not associated with a reduced risk of either SARS-CoV-2 infection or hospitalization due to COVID-19 [75]. With respect to adverse effects, prior usage of NSAIDs has been reported to significantly increase the risk of stroke in COVID-19 patients (OR = 2.32, 95% CI 1.04–5.2, I2 = 0%), whilst not thrombotic events (including deep vein thrombosis, pulmonary embolism, myocardial infarction) or renal failure [75].

### 3.3. Ketoprofen Lysine Salt in the Therapeutic Scenario of SARS-CoV-2 Infection

#### 3.3.1. Ketoprofen Lysine Salt Mechanism of Action

An optimal NSAID should give relief and prevent complications and worsening of the disease. Ketoprofen is a NSAID belonging to the family of propionic derivates, with analgesic, anti-inflammatory, antipyretic, and antiplatelet properties [76,77].

Salification of ketoprofen with the lysine amino acid allows for higher solubility that facilitates a more rapid and complete absorption of the drug with a high peak plasma concentration reached after 15 min vs. 60 min [78,79,80,81]. Ketoprofen lysine salt (KLS) is a non-selective NSAID but also demonstrates high activity on COX-2 and inhibits the lipoxygenase pathway of the arachidonic acid cascade leading to a decrease in the synthesis of leukotrienes [82]. Ketoprofen, in common with other NSAIDs, has both peripheral and central sites of action [75] through the inhibition of both nitric oxide (NO) and COX synthase in the brain [83] and is rapidly and readily distributed into the central nervous system passing the blood brain barrier within 15 min, thanks to its high level of liposolubility (Figure 1) [79]. KLS penetrates extensively into the tonsillar tissue and in the adipose tissue; this is particularly interesting since recent data demonstrated the ability of SARS-CoV-2 to infect human adipocytes and that subjects with severe COVID-19 have higher local visceral adipose tissue inflammation [84]. Moreover, the use of ketoprofen lysine salt is associated with low prevalence of gastrointestinal adverse effects due to the improved solubility and bioavailability that determine a better gastrointestinal safety profile [80]. Compared with ibuprofen, ketoprofen lysine salt has minor effects on the gastric mucosa and therefore a better gastrointestinal safety profile; probably due to the protective effect exerted by lysine in the ketoprofen molecule, for its antioxidant effect and for its capacity to stimulate the production of endogenous gastro-protective proteins [85,86].

Several clinical studies and meta-analysis have demonstrated that ketoprofen lysine salt provided better and longer analgesic control as well as greater and faster pain relief than other NSAIDs, showing that it has the highest ratio between antinflammatory and analgesic effect, being the best molecule among the different NSAIDs compared, such as ibuprofen and diclofenac [78,87]. This strong efficacy has been confirmed in a recent meta-analysis underlying the better analgesic effect of ketoprofen compared with ibuprofen and diclofenac [88,89]. Another interesting effect of ketoprofen lysine salt is its role in reducing platelet aggregation and the synthesis of thromboxane B_2_ with a potency superimposable to that of aspirin [87,90]. This is particularly important in the management of COVID-19 disease as during the cytokine storm vessels’ endothelium is activated reducing prostacyclin and NO production, important antiaggregant mediators, causing diffuse coagulopathy as part of the systemic inflammatory response syndrome [91]. For this reason, in the early phases of the SARS-CoV-2 infection it is particularly important to choose NSAIDs with antiplatelet activity. This effect should be considered as potentially differentiating in the choice of an antinflammatory drug, since not all NSAIDs share this antiaggregant activity, for example ibuprofen has a negligible effect [90,92]. Moreover, another important aspect that should be considered when treating patients with COVID-19 is background cardioprotective therapy with low-dose aspirin. For these patients, the appropriate choice of NSAIDs should be carefully evaluated, since pharmacodynamic interactions between NSAIDs and low-dose aspirin are not classified as class effect, because not all NSAIDs interact with aspirin to the same extent: ketoprofen lysine salt does not interfere with antiplatelet activity, while clinical dosing regimen of ibuprofen or naproxen competitively inhibit aspirin’s antiplatelet effect [93,94,95]. Altogether, while the above-reported pharmacokinetic and pharmacodynamic proprieties of KLS suggest potential for use in COVID-19 patients, at this time no significant evidence from studies in COVID-19 are currently available.

#### 3.3.2. Ketoprofen Lysine Salt Cardiovascular Safety

In the last two years, real life studies and clinical experience have clarified that SARS-CoV-2 can impair cardiovascular system infecting heart and vascular tissues via ACE2 (angiotensin-converting enzyme 2), highlighting that cardiovascular diseases highly influence the susceptibility to and the outcomes of SARS-CoV-2 infection.

The systemic inflammation related to COVID-19 could accelerate the worsening of subclinical disorders or cause de novo cardiovascular diseases [96,97] such as myocardial injury or acute coronary syndrome largely linked to advanced systemic inflammation. Attention should be paid to the potential drug–disease interactions, preexisting cardiovascular diseases and drug cardiovascular safety profile in COVID-19 patients [98]. Some non-selective NSAIDs could increase the cardiovascular (CV) risk [99] and actually a study to investigate CV safety in NSAIDs funded by the European Commission (EMA) is still in progress [100,101]. Among the selected NSAIDs, the cardiovascular risk increased for seven traditional NSAIDs (diclofenac, ibuprofen, indomethacin, ketorolac, naproxen, nimesulide, and piroxicam) while ketoprofen showed the lowest relative risk for heart failure (OR 1.04 (0.96–1.12) and for acute myocardial infarction (OR_pool_ 1.00 (0.86 to 1.16) [100,101], showing a cardiovascular safety profile better than many other NSAIDs. Furthermore, a recent study applying a very elegant in vitro model, strongly supports the hypothesis that immortalized human cardiomyocytes exposed to ketoprofen are subjected to tolerable stress events while diclofenac exposition leads to cell death, dramatically increasing ROS (reactive oxygen species) production. Increased ROS levels cause proteasome dysfunction and the opening of mitochondrial permeability transition pores (mPTPs) allowing the release of cytochrome c and the activation of caspase-3/9, thus inducing intrinsic apoptotic pathway. This study shows an alteration of mPTP only in cells exposed to diclofenac; conversely cells exposed to ketoprofen show the same behavior of untreated cells [102].

## 4. Future Perspective

Inflammation represents a key factor of SARS-CoV-2 infection and a growing body of literature is focusing on this issue. By exploiting their anti-inflammatory proprieties, NSAIDs have been regarded as safe drugs to be utilized in COVID-19. However, new approaches, from diagnosis to treatment, are currently under the spotlight to tackle this issue. Geromichalou et al. have investigated the potential antiviral activity of copper (II) complexes with non-steroidal anti-inflammatory drugs on various SARS-CoV-2 proteins [72]. Although in vitro model, authors demonstrated that these molecular complexes may potentially impact on SARS-CoV-2 replication [72]. Nanomaterials and nanotechnologies may play a significant role not only for a quick and accurate diagnosis; they also may improve COVID-19 treatment as well as new strategies for vaccines [103,104].

## 5. Conclusions

Targeting pro-inflammatory pathways in early phases of COVID-19 is one of the key strategies in avoiding progression of the disease. Current literature should however be interpreted with caution as the research has been conducted during different COVID-19 waves and the presence of SARS-CoV-2 variants is not negligible. However, NSAIDs, because of their interaction with different pro-inflammatory mediators, may play pleiotropic effects in COVID-19. Available evidence about the role of NSAIDs in SARS-CoV-2 are based on retrospective studies. In this respect, large prospective randomized clinical trials are needed to better investigate the efficacy and safety of these drugs also exploring the different pharmacokinetic properties existing among the drugs. Finally, the evolution of therapeutic strategies for the early stages of SARS-CoV-2 infection should be oriented in developing combination of complementary drugs able to interact with this complex network between virus, host cell, and innate immune cells.

## Figures and Tables

**Figure 1 molecules-27-08919-f001:**
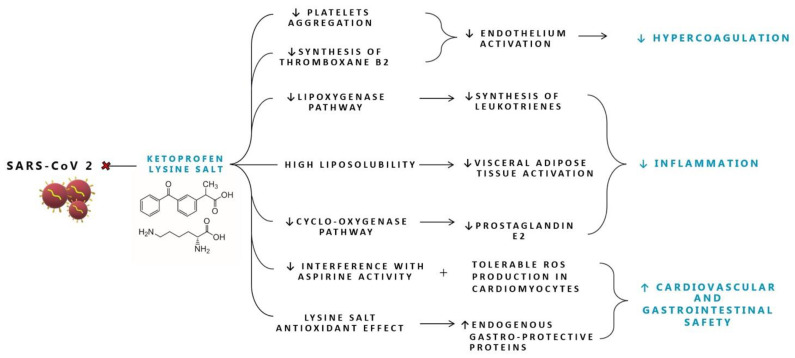
Proposed mechanisms of action of Ketoprofen lysine salt in SARS-CoV-2 infection. ROS: reactive oxygen species.

## Data Availability

Not applicable.

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
