# Peer review of "Strategies Tackling Viral Replication and Inflammatory Pathways as Early Pharmacological Treatment for SARS-CoV-2 Infection: Any Potential Role for Ketoprofen Lysine Salt?"

_molecules, 2022, doi:10.3390/molecules27248919_

Round 1

Reviewer 1 Report

Introduction provides a sound rationale for the review.  Error in line 33 (bleed) should be blood

Section 1.1 is a thorough review and well  written with minor errors.

Beginning with section 2.1.1.1 There were numerous English usage errors.

Line 71 spike-protein should be spike protein

Line 72 should be dimer

Line 130 “has deceased”

Line 136 remove “has”

Line 140 replace numeral “1” with the word one

Line 147 replace “has” with was

2.1.1.5 The term “microorganisms” should not be used as a comparison because viruses are not considered microorganisms.  Line 163

Several abbreviations were used without previously stating the term(s):  line 213 HFNC; line 214 PO; line 291 NIH; line 331 EMA

Line 166 “diffused” Usage error

Line 173 “protein” should be proteins

Line 174 replace numeral “6” with the word six

Line 177 “exploited” usage error

2.14 Extensive errors – awkward sentences; revisions required (lines 231-232; 234-236; line 238 “Mostly of virus” ???; lines 240-243

Line 248 “are” should be were

Lines 251-252 “have been needed to”

Line 259  used singular “a” with plural effects; “due its” should be due to its

Line 263 “due” should be due to

Line 271 should not use first person “we have”

Line 293 suggest removing “also” 

Line 296 replace “on” with of

Line 303 “administrated” replace with administered

Line 306 tense error “are”; should be were

Line 313 tense error “test” 

Line 315 “reducing need” suggest change to reducing the need

Line 332 “well-indicated use” – meaning not clear

Line 350 Should revise “In the last years, their prescription has decreased” usage error

Line 360 tense error “dampen” should be past tense

I have major objections to Figure 1. Was Figure 1 taken from reference 66 or did authors create this figure from other studies?  It could not be based on data in reference 66 alone yet no other reference in the manuscript is made to this figure.  The proposed response to SARS-CoV-2 infection is speculative.  It is not based on any experimental studies, yet authors do not indicate that this is a proposed response.  

Line 404 “provide” should be …salt provides or …salt provided

Line 408 “underling” – should be underlying

Lines 457-458 should revise to make it more

Title does not accurately reflect the content of the paper.  There is minimal inclusion of research on Ketoprofen lysine salt since there are not research studies that focus on response to or impact of Ketoprofen lysine salt to inflammation induced by any virus infection.  I don’t feel a review should go beyond and propose roles not supported by research. The conclusion is speculative and an over-reach.

Author Response

Reviewer 1

Introduction provides a sound rationale for the review.  Error in line 33 (bleed) should be blood

Section 1.1 is a thorough review and well  written with minor errors.

Beginning with section 2.1.1.1 There were numerous English usage errors.

Line 71 spike-protein should be spike protein

Line 72 should be dimer

Line 130 “has deceased”

Line 136 remove “has”

Line 140 replace numeral “1” with the word one

Line 147 replace “has” with was

  • We thank the referee for the above-reported comments. All the English errors, mistakes and typos have been corrected

2.1.1.5 The term “microorganisms” should not be used as a comparison because viruses are not considered microorganisms.  Line 163

  • We thank the referee for the comment. We have removed this inaccurate term.

Several abbreviations were used without previously stating the term(s):  line 213 HFNC; line 214 PO; line

291 NIH; line 331 EMA

  • We thank the referee for this comment. We are very sorry for this inaccuracy. We have appropriately reported the acronyms in the extended version when introduced in the manuscript.

Line 166 “diffused” Usage error

Line 173 “protein” should be proteins

Line 174 replace numeral “6” with the word six

Line 177 “exploited” usage error

  • We thank the referee for the above-reported comments. All the English errors, mistakes and typos have been corrected

2.14 Extensive errors – awkward sentences; revisions required (lines 231-232; 234-236; line 238 “Mostly of virus” ???; lines 240-243

  • We thank the referee for this comment. This part have been completely rephrased.

Line 248 “are” should be were

Lines 251-252 “have been needed to”

Line 259  used singular “a” with plural effects; “due its” should be due to its

Line 263 “due” should be due to

Line 271 should not use first person “we have”

Line 293 suggest removing “also” 

Line 296 replace “on” with of

Line 303 “administrated” replace with administered

Line 306 tense error “are”; should be were

Line 313 tense error “test” 

Line 315 “reducing need” suggest change to reducing the need

Line 332 “well-indicated use” – meaning not clear

Line 350 Should revise “In the last years, their prescription has decreased” usage error

Line 360 tense error “dampen” should be past tense

I have major objections to Figure 1. Was Figure 1 taken from reference 66 or did authors create this figure from other studies?  It could not be based on data in reference 66 alone yet no other reference in the manuscript is made to this figure.  The proposed response to SARS-CoV-2 infection is speculative.  It is not based on any experimental studies, yet authors do not indicate that this is a proposed response. 

  • We really want to thank the referee for this comment. We do understand that the information should be transferred to readers with particular attention based on possible misinterpretation. The Figure was made by the Authors as a summary of all the pharmacological evidences described in the paragraph – from studies not in COVID19 – to offer a potential perspective for use in COVID-19 which however should be confirmed in clinical trials. For these reasons, we have changed the Figure title highlighting that this is a proposed mechanism and we added a sentence to emphasize the absence of data available from COVID-19  “Altogether, while the above-reported pharmacokinetic and pharmacodynamic proprieties of KLS suggest potential for use in COVID-19 patients, at this time no significant evidence from studies in COVID-19 is currently available”. Once again, many thanks for this valuable comment

Line 404 “provide” should be …salt provides or …salt provided

Line 408 “underling” – should be underlying

Lines 457-458 should revise to make it more

  • We thank the referee for the above-reported comments. All the English errors, mistakes and typos have been corrected

Title does not accurately reflect the content of the paper.  There is minimal inclusion of research on Ketoprofen lysine salt since there are not research studies that focus on response to or impact of Ketoprofen lysine salt to inflammation induced by any virus infection.  I don’t feel a review should go beyond and propose roles not supported by research. The conclusion is speculative and an over-reach.

  • We thank the referee this valuable comment. In this review, Authors aim to describe the state of art about early treatment of COVID-19. Evidence suggests that NSAIDs might play a relevant role as part of the treatment along with other agents approved from regulatory agencies though we tried to balance all the available information. Furthermore, while different NSAIDs are currently available with distinctive pharmacokinetic features, these Authors would offer a perspective in use of KLS. We do acknowledge to this referee the absence of current studies in this field and we add this important limitation in the paragraphs and in the conclusion as well. Also, we have tried to avoid unbalanced information and any speculative comments in the final lines which have been completely rephrased. Finally, we have slightly changed the title of the manuscript to avoid any misinterpretation from the readers. Once again, many thanks for this comment.

Reviewer 2 Report

1.It is suggested to add new research studies in introduction.
2.what is the suggestion of this study for future works?
3.Please discuss about the using of new biomaterials with new technologies including nanomedicine.
4.It will be better to add the role of mitochondria and gap junction proteins. 
5.Please add these references for your discussion part of manuscript and bold your study novelty :

DOI:10.2217/nnm-2020-0441

DOI: 10.3390/microorganisms9020232

DOI:10.1016/j.jinorgbio.2022.111805

Author Response

1.It is suggested to add new research studies in introduction

  • We thank the referee for this comment. In the revised version of the manuscript the references have been updated and extended

2.what is the suggestion of this study for future works?

  • We thank the referee for this comment. In the conclusion, we have added this perspective suggesting that current limitation of the available literature of this field. Also, any substantial comparison among agents has been reported at this time. We included these comments in the revised version of the manuscript. Many thanks for this valuable consideration.

3.Please discuss about the using of new biomaterials with new technologies including nanomedicine.

  • We thank the referee for this comment. We have now included this topic in our review in a dedicated paragraph.

4.It will be better to add the role of mitochondria and gap junction proteins. 

  • We thank the referee for the comment. We have now extended these aspects in the revised version of the manuscript

5.Please add these references for your discussion part of manuscript and bold your study novelty :

DOI:10.2217/nnm-2020-0441

DOI: 10.3390/microorganisms9020232

DOI:10.1016/j.jinorgbio.2022.111805

  • We thank the referee for this comment. The references have been updated including these above-reported scientific articles which we believe are very important for the readers. Once again, many thanks for this valuable revision of our manuscript.

Round 2

Reviewer 1 Report

This version is much improved.  I have only two corrections:

Line 67 – should be “by killing…”

Line 132  Simplicity is often better.  Again “have deceased” is a usage error.  Suggest simply state “no patients died …”

Author Response

We thank the referee for the above-reported comments. All the English errors, mistakes and typos have been corrected